# Exploring the Mechanism of 2,4-Dichlorophenoxyacetic Acid in Causing Neurodegenerative Diseases Based on Network Toxicology and Molecular Docking

**DOI:** 10.3390/ijms262411980

**Published:** 2025-12-12

**Authors:** Yucheng Yan, Xiaoqi Luo, Yanan Song, Haoxuan Gao, Yuwen Wang, Yiman Li, Huifang Yang, Jian Zhou

**Affiliations:** 1Department of Occupational and Environmental Health, School of Public Health and Management, Ningxia Medical University, Yinchuan 750004, China; yanyucheng0315@163.com (Y.Y.); luoxiaoqi0121@163.com (X.L.); s759784159@163.com (Y.S.); haoxuangao1997@163.com (H.G.); wildwyw@163.com (Y.W.); 15517793636@163.com (Y.L.); 2Key Laboratory of Environmental Factors and Chronic Disease Control, Yinchuan 750004, China

**Keywords:** 2,4-Dichlorophenoxyacetic acid, neurodegenerative diseases, network toxicology, molecular docking

## Abstract

This study employed an integrated network toxicology and molecular docking approach to explore the molecular mechanisms by which the herbicide 2,4-Dichlorophenoxyacetic acid (2,4-D) may contribute to neurodegenerative diseases (NDDs). We identified 89 common targets through the intersection of potential 2,4-D-related targets and NDD-associated genes. Among these, 12 core targets—including NFKB1, PPARG, SERPINE1, NOS3, and NFE2L2—were highlighted via protein–protein interaction network analysis. Functional enrichment revealed that these targets are involved in key pathways such as inflammatory response, oxidative stress, metabolic dysregulation, and synaptic dysfunction. Molecular docking further confirmed strong binding affinities between 2,4-D and all core targets (binding energy ≤ −5.1 kcal·mol^−1^). These findings systematically reveal, for the first time, a multi-target and multi-pathway mechanism through which 2,4-D may induce neuronal injury, providing a theoretical basis for assessing environmental risk in neurodegeneration.

## 1. Introduction

2,4-Dichlorophenoxyacetic acid (2,4-D) is a widely used herbicide that persists in the environment and has been linked to adverse effects in wildlife and humans [1,2]. Residual 2,4-D has been detected in human urine [3], and exposure is associated with neurobehavioral alterations in both animal models [4] and human populations, including the elderly [5], indicating its potential neurotoxicity.

Neurodegenerative diseases are a growing global public health concern, and environmental chemical exposures, including pesticides, are recognised as potential risk factors [6]. Notably, epidemiological studies have identified a significant association between 2,4-D exposure and neurodegenerative disease risk, particularly among agricultural workers [7]. The lipophilic nature of 2,4-D, conferred by its chlorinated benzene ring, may facilitate blood–brain barrier penetration and accumulation in neural tissues, similar to organochlorine pesticides. Exposure to Agent Orange, which contains 2,4-D, has been linked to brain tissue damage, biochemical abnormalities, and increased incidence of Parkinson’s disease [8,9]. Further support comes from cohort studies indicating that long-term pesticide exposure, including 2,4-D, heightens the risk of neurodegeneration [10].

To elucidate the underlying molecular mechanisms, this study integrates network toxicology—which constructs multi-level networks linking compounds, targets, and pathways—and molecular docking, a computational approach to predict binding interactions [11]. By building a regulatory network between 2,4-D and neurodegenerative disease-related targets, we aim to systematically explore its multi-target and multi-pathway toxicity mechanisms. This work seeks to provide molecular insights into 2,4-D-induced neurodegeneration and support risk assessment of its neurotoxic effects.

## 2. Results

### 2.1. Screening of Common Targets

Potential protein targets of 2,4-D were predicted using the Pharm Mapper and SwissTargetPrediction databases, yielding 119 candidates. Meanwhile, 11,637 targets associated with neurodegenerative diseases were retrieved from the OMIM, GeneCards, and TTD databases. The figure shows the two-dimensional structure of 2,4-D (Figure 1A). The intersection of these two sets identified 89 common target genes (Figure 1B), which formed the basis for subsequent analysis of the molecular mechanisms underlying 2,4-D-induced neurotoxicity.

### 2.2. Construction of PPI Network and Identification of Core Targets

A protein–protein interaction (PPI) network was constructed for the 89 common targets using the STRING database, resulting in a network comprising 77 nodes and 168 edges (Figure 1C). Topological analysis was performed in Cytoscape 3.10.0, and based on a comprehensive evaluation of three centrality metrics—degree, betweenness, and closeness—12 hub genes were identified as core targets: NFKB1, PPARG, SERPINE1, PPARA, NOS3, NFE2L2, SLC2A1, CSNK2A1, CSNK2A2, PTGS1, TOP1, and CA9 (Figure 1D,E). Among these, NFKB1 occupied a central position within the network.

### 2.3. Gene Ontology (GO) Enrichment Analysis

GO enrichment analysis of the core targets revealed significant associations across three categories. Under Biological Process, terms such as organic acid transport, response to oxygen levels, smooth muscle cell migration, and transmission of nerve impulses were enriched, suggesting that 2,4-D may disrupt neurometabolic balance, vascular function, and signal transduction. For Cellular Component, significant enrichment was observed for synaptic membrane, vesicle lumen, and voltage-gated channel complexes, indicating potential effects on synaptic structure and vesicular transport systems. In Molecular Function, the targets were primarily associated with sodium ion transport, nuclear receptor activity, and carboxylate binding, revealing possible disturbances in ion homeostasis and transcriptional regulation at the molecular level (Figure 2A–D).

### 2.4. KEGG Pathway Enrichment Analysis

KEGG pathway analysis identified 33 significantly enriched pathways. The ten most relevant pathways were grouped into four key themes: (1) inflammation and immunity, including complement and coagulation cascades and efferocytosis; (2) metabolic dysregulation, such as insulin resistance and adipocytokine signaling pathway; (3) vascular and oxidative damage, exemplified by the AGE-RAGE signaling pathway in diabetic complications and fluid shear stress and atherosclerosis; and (4) neurological dysfunction, involving the synaptic vesicle cycle and neuroactive ligand–receptor interaction (Figure 2E). These results systematically indicate that 2,4-D may exert its toxicity by disrupting an integrated network encompassing inflammatory, metabolic, vascular, and neural functions.

### 2.5. Molecular Docking of 2,4-D with Core Targets of Neurodegenerative Diseases

To verify the potential direct interaction between 2,4-D and the core targets, molecular docking was performed using CB-Dock2. The results demonstrated that 2,4-D could stably bind to the predicted binding pockets of all 12 core targets (representative structures shown in Figure 3A,B). The calculated binding energies ranged from −5.1 to −7.1 kcal/mol (Table 1), with all values below −5.1 kcal/mol, indicating strong theoretical binding affinity. These computational findings support the plausibility of direct interactions between 2,4-D and the identified core targets.

## 3. Discussion

This study primarily utilises toxicological database analysis and literature review to elucidate the mechanism of action of 2,4-D in causing neurodegenerative diseases. As one of the most widely used herbicides globally in agriculture, gaining a thorough understanding of its toxicological effects is crucial for preventing and mitigating its impact on human health. This study identified a total of 119 target sites for 2,4-D using the Pharm Mapper and Swiss Target Prediction databases, with 89 of these target sites overlapping with genes associated with neurodegenerative diseases. Subsequently, through the PPI network, 12 core targets with the highest specificity for neurodegenerative diseases were identified: NFKB1, PPARG, SERPINE1, PPARA, NOS3, NFE2L2, SLC2A1, CSNK2A2, CSNK2A1, PLAU, PTGS1, TOP1, and CA9. Next, the core targets were subjected to GO and KEGG enrichment analysis using the SRplot platform. Finally, molecular docking was performed on the aforementioned core targets to explore the molecular interactions between 2,4-D and the core targets, and to demonstrate the binding energy between 2,4-D and the targets, thereby elucidating the potential molecular mechanism of 2,4-D-induced neurodegenerative diseases. Therefore, 2,4-D directly binds to 12 core target proteins, disrupts 10 neurodegenerative disease-related pathways, and ultimately causes neuronal damage.

Analysis of core inflammatory gene targets reveals that NFKB1, a member of the NF-κB/Rel protein family, functions as a “molecular switch” in neuroinflammation. Molecular docking in this study demonstrated a binding energy of −5.1 kcal·mol^−1^ between 2,4-D and NFKB1, indicating potential direct interference with NF-κB signal transduction. The predicted 2,4-D binding sites in this study have important functional implications. ASN-102, THR-101, GLN-203, and HIS-107 are all located in the Rel homologous domain of NFKB1, which is directly responsible for recognising and binding to the kappa B sequence on DNA. Especially, GLN-203 and HIS-107 are highly conserved in various species and are key residues for protein DNA base-specific hydrogen bonding (Appendix A). Significantly reduced NFKB1 expression has been observed in Parkinson’s disease patient samples, and NFKB1 knockout (NFKB1^−/−^) mice exhibit exacerbated microglial activation and dopaminergic neuron loss [12]. These findings strongly align with our conclusion that 2,4-D-mediated inhibition of NFKB1 function leads to impaired efferocytosis (apoptotic cell clearance). An Y et al. [13] demonstrated that insulin and lipoxin A4 treatment reversed osteoclast (OC)-mediated bone resorption in vitro, highlighting the role of NF-κB in efferocytosis within osteoclasts. Similarly, 2,4-D may obstruct inflammatory resolution via an analogous mechanism, thereby fostering a chronic neuroinflammatory microenvironment. Consistent with our molecular docking results, 2,4-D targets NFKB1, significantly disrupting efferocytosis. This impairment in cellular clearance contributes to inflammatory accumulation and the initiation of neurodegenerative diseases (NDDs). PTGS1 (cyclooxygenase-1, COX-1), the rate-limiting enzyme in prostaglandin synthesis, exhibits high binding affinity with 2,4-D (−6.8 kcal·mol^−1^), suggesting dual pathological implications. Wang Y et al. [14] investigated the inflammatory response of PTGS1 (COX-1) using TNFα treatment, reporting that PTGS1 upregulation ameliorated the response to inflammatory cytokines. Moreover, PTGS1 genetic variants impair platelet aggregation and function and are closely associated with severe bleeding complications [15,16]. Additionally, it mediates neuroinflammation and oxidative response, and its upregulated expression in NDDs models and the protective effect of inhibitors have been experimentally supported [17]. Follow-up studies further identified a significant association between the PTGS1 locus and ischemic stroke (IS) recurrence in Chinese patients [18]. In concordance with our findings, 2,4-D targets PTGS1, consequently interfering with the complement and coagulation cascades. This interference, potentially via coagulation-complement cross-activation, impacts the pathophysiological mechanisms underlying NDDs. SERPINE1 (plasminogen activator inhibitor-1, PAI-1) primarily regulates coagulation and fibrosis. Studies have shown that SERPINE1 binds to p65, facilitating its nuclear translocation and thereby activating the NF-κB inflammatory signaling pathway [19]. The strong binding affinity observed between 2,4-D and SERPINE1 in this study (−6.2 kcal·mol^−1^) may potentiate this effect, promoting a prothrombotic state and cerebral microcirculatory dysfunction. Relevant experimental studies confirmed that serpine1 expression levels were elevated in mouse models and significantly upregulated in the coronary artery disease (CAL) cohort [20], providing molecular substantiation for 2,4-D-induced cerebrovascular complications. This evidence suggests that SERPINE1, by inhibiting fibrinolysis and interacting with the complement and coagulation cascades, promotes thrombosis, disrupts cerebral microcirculation, and contributes to cerebrovascular injury and cognitive decline.

Analysis of metabolic regulatory gene targets identifies PPARG (PPARγ) and PPARA (PPARα) as core peroxisome proliferator-activated receptors governing glucose and lipid metabolism. This study demonstrates that 2,4-D targets both PPARG and PPPARA with binding affinities of −5.9 kcal·mol^−1^ and −6.1 kcal·mol^−1^, respectively. Regarding targets associated with metabolic dysregulation, PPARG and PPARα are members of the peroxisome proliferator-activated receptor (PPAR) subfamily. PPARG expression is upregulated in atherosclerosis (AS) mouse models compared to normal controls [20]. Furthermore, reactive oxygen species (ROS) accumulation is closely linked to PPARG signaling, and PPARG agonists significantly ameliorate the pathological manifestations of insulin resistance [21]. As a transcription factor, PPARA-mediated autophagy influences Alzheimer’s disease (AD) pathology, and PPARA agonists can reverse memory deficits and anxiety-like behaviors in experimental mice. Relevant experimental studies have confirmed that ppar-γ regulates microglia activation/polarisation and subsequent neuroinflammation/oxidative stress through hmgb1/nf-κ B and nrf2/keap1 signaling pathways [22]. Consistent with these findings, our results reveal that 2,4-D targets PPARG/PPARA, leading to metabolic dysregulation. This disruption impacts insulin resistance and adipocytokine signaling pathways, thereby promoting the development of neurodegenerative diseases. SLC2A1 (Glucose Transporter 1, Glut1), highly expressed at the blood–brain barrier (BBB) and essential for central nervous system (CNS) function, facilitates glucose transport into the brain. Dysfunction of this transporter precipitates an energy crisis. In this study, the binding affinity between 2,4-D and SLC2A1 was determined to be −6.0 kcal·mol^−1^. Zebrafish SLC2A1 knockdown models exhibit brain endothelial cell loss and BBB leakage, resulting in vasogenic cerebral edema [23]. Additionally, this dysfunction can trigger abnormal endothelial cell excitability, leading to CNS inflammation and neuronal loss [24]. Our findings indicate that 2,4-D-associated targeting of SLC2A1 contributes to cerebral energy metabolism impairment. Reduced glucose uptake accelerates cognitive decline, corroborating previous research. By inhibiting insulin signaling and impairing cerebral glucose uptake, 2,4-D likely induces memory dysfunction. This metabolic compromise may potentiate the accumulation of pathological proteins (Aβ, tau, α-synuclein), ultimately inducing the onset of neurodegenerative diseases (NDDs).

Starting from gene targets associated with oxidative stress and vascular injury, endothelial nitric oxide synthase (NOS3) primarily catalyses the production of nitric oxide (NO), thereby regulating vasodilation and exerting antioxidant effects. In this study, 2,4-D demonstrated the highest binding affinity for NOS3 (−7.1 kcal·mol^−1^), indicating its potential to cause vascular dysfunction: specifically, reduced cerebral blood flow perfusion, promotion of microthrombus formation, aggravation of oxidative stress, and endothelial cell senescence. In this study, we found that 2,4-D specifically binds to the hydrophobic pocket composed of TRP-445, ALA-446, and TRP-447 on NOS3 and forms a hydrogen bond with ser-102. This binding region has important functional significance: it highly overlaps with the key role of endogenous calmodulin (CAM) in mediating DAPK1 activity (Appendix A) [25]. More importantly, the hydrophobic aromatic characteristics of this site make it easy to be targeted by halogenated aromatic environmental pollutants. For example, polychlorinated biphenyls with similar structures have been reported to interfere with protein function by binding to similar interfaces [26]. In vitro cellular models have revealed that NOS3 promotes angiogenesis potential [27]. Austin et al. used NOS3 knockout mice to cross with Alzheimer’s disease model mice. The cognitive function was evaluated by the Morris water maze, and the changes in cerebral blood flow, and tau phosphorylation were detected by laser speckle contrast imaging and Western blot, respectively, so as to prove that eNOS deficiency exacerbated Alzheimer’s disease pathology [28]. Case–control studies genotyping the common structural polymorphism Glu/Asp at codon 298 of the *NOS3* gene suggest that NOS3 may represent an important genetic risk factor for Alzheimer’s disease (AD) [29]. Consistent with these findings, we hypothesise that 2,4-D inhibits endothelial nitric oxide synthase, leading to reduced NO production and impaired vasodilatory function. Furthermore, it is proposed that 2,4-D promotes cerebrovascular lesions via fluid shear stress and atherosclerosis-related pathways, which are recognised vascular factors contributing to AD, thereby facilitating the onset and progression of neurodegenerative diseases (NDDs). Nuclear factor erythroid 2-related factor 2 (NFE2L2) serves as the master regulator of the antioxidant response, wherein oxidative stress contributes to neuronal loss. The binding of 2,4-D to NFE2L2 (−5.2 kcal·mol^−1^) is predicted to impair the activation of antioxidant response elements (AREs). NFE2L2 confers neuroprotective effects at both the photoreceptor and retinal ganglion cell levels [30]. Moreover, its dysfunction is closely linked to neurodegenerative diseases, including AD, Parkinson’s disease (PD), and amyotrophic lateral sclerosis. Additionally, NFE2L2 activators have demonstrated therapeutic efficacy against NDDs in both animal models and human cells [31]. In alignment with the present study, these observations imply that 2,4-D disrupts the antioxidant function of NFE2L2, thereby exacerbating oxidative damage induced by the AGE-RAGE signaling pathway in diabetic complications, which ultimately leads to dopaminergic neuron death and blood–brain barrier disruption. Carbonic anhydrase 9 (CA9) primarily functions in regulating pH balance and mediating responses to hypoxia, and is highly expressed under hypoxic conditions. The interaction between CA9 and 2,4-D (−6.4 kcal·mol^−1^) may induce astrocyte damage by disrupting pH homeostasis. Gene Ontology (GO) analysis identifies CA9 as a potential diagnostic biomarker for AD [32], while knockdown of CA9 has been shown to mitigate lipid peroxidation in glioma cells [33]. This evidence suggests that hypoxia may induce CA9 expression, resulting in cerebral acidosis and astrocyte damage, which consequently promotes neuropathological changes in the brain.

Focusing on synaptic function-related gene targets, casein kinase 2 subunits alpha and alpha’ (CSNK2A1/CSNK2A2) function to phosphorylate pathological proteins such as tau and α-synuclein. In this study, 2,4-D targeted these kinases with binding energies of −6.5 kcal·mol^−1^ (for CSNK2A1) and −6.6 kcal·mol^−1^ (for CSNK2A2), suggesting a potential mechanism for synaptic impairment: Studies utilising a Drosophila model demonstrated that CSNK2A1 regulates DNA damage in both Drosophila and human stem cell-derived neural progenitor cells, identifying potential strategies to ameliorate Alzheimer’s disease (AD)-associated neurodegeneration [34]. Furthermore, CSNK2A2, validated by ELISA assays, is implicated in neuronal activities and exhibits a strong association with the pathogenesis of neurodegenerative diseases [35]. Collectively, these findings align with our results and support the hypothesis that 2,4-D, acting via CSNK2A1/CSNK2A2, perturbs the synaptic vesicle cycle and neuroactive ligand–receptor interactions. This perturbation may dysregulate the phosphorylation of synaptic vesicle-associated proteins, inhibit neurotransmitter release, and ultimately contribute to the development of neurodegenerative diseases (NDDs). DNA topoisomerase I (TOP1) primarily regulates DNA unwinding and is involved in transcriptional repair, underscoring the critical link between DNA unwinding function and neuronal DNA repair. Hippocampal dysfunction adversely impacts cognitive functions such as learning and memory, with severe cases substantially diminishing quality of life. TOP1 has been implicated in sleep deprivation (SD)-induced microglial activation and neuronal damage; specifically, TOP1 mediates microglial activation, contributing to SD-induced hippocampal neuronal damage and behavioral deficits [36]. Consistent with our findings, this suggests that 2,4-D may induce TOP1 dysfunction, resulting in neuronal DNA damage, disruption of associated pathways, and consequent neurological dysfunction.

In our study, we utilised internet data and employed network toxicology and molecular docking methods to investigate the mechanism of action of 2,4-D in causing neurodegenerative diseases. We identified the first 12 gene targets. Therefore, patients with neurodegenerative diseases may experience corresponding changes in the expression levels of these 12 genes, and their differential expression may further accelerate the progression of the disease. That is, 2,4-D directly or indirectly induces toxicity through core genes, which then regulate the 10 most significantly associated pathways. Ultimately, pathway dysregulation leads to the development of neurodegenerative disease pathology. To systematically elucidate the multi-faceted roles of the predicted core targets, their associated pathways, and primary biological functions are summarised in Table 2.

In summary, this study proposes an integrative mechanistic hypothesis: as an environmental neurotoxin, 2,4-D simultaneously targets multiple key regulatory proteins such as NFKB1, NOS3, and NRF2, cooperatively inducing chronic dysregulation of four major pathological networks: “neuroinflammation–oxidative stress–vascular injury–metabolic dyshomeostasis.” These networks intertwine and amplify each other through positive feedback loops, ultimately disrupting neuronal homeostasis, synaptic integrity, and neurovascular unit function. This constitutes the potential molecular and systems-level biological basis through which 2,4-D exposure may promote the onset and progression of neurodegenerative diseases. This framework provides a clear theoretical direction for subsequent research and risk assessment (Figure 4).

### Limitations and Future Perspectives

It is important to note that the conclusions of this study are entirely based on computational models and bioinformatic analyses, which have inherent limitations. The results from network toxicology and molecular docking are theoretical predictions and may not fully replicate the complex dynamic environment, compound metabolism, tissue specificity, and integrated physiological regulatory networks within a living organism.

Furthermore, the research methodology itself may introduce selection bias. The identification of core targets relies on specific databases (e.g., SwissTargetPrediction, GeneCards) and the set of network topology parameters (such as Degree ≥ 2). The molecular docking results are also constrained by the scoring function of the chosen tool (CB-Dock2). While these choices follow standard procedures, they may still influence the direction of the final findings. Therefore, the mechanistic hypotheses proposed in this paper require further rigorous validation through wet-lab experiments (e.g., binding assays, functional verification).

## 4. Materials and Methods

### 4.1. Identification of Potential Targets for 2,4-D

The SMILES sequence of 2,4-D ([H]OC(=O)C([H])([H])OC1=C(Cl)C([H])=C(Cl)C([H])=C1[H]) was retrieved from PubChem (PubChem CID 1486) https://pubchem.ncbi.nlm.nih.gov (accessed on 20 October 2025), followed by downloading its 2D structure (SDF format) and 3D-optimised conformation. The SMILES data were uploaded to Swiss Target Prediction https://swisstargetprediction.ch/ (accessed on 20 October 2025) with species set to Homo sapiens and target selection to “All targets”. Predicted targets with Probability >0.1 were downloaded and saved as a CSV file. Separately, “2,4-Dichlorophenoxyacetic acid” was queried in SuperPred to filter targets with Model accuracy >70%. Results from both platforms were integrated, duplicates removed, and a comprehensive 2,4-D target library constructed.

### 4.2. Screening of Neurodegenerative Disease-Associated Targets

Identification of the medical subject term “Neurodegenerative diseases” from the MeSH database https://meshb.nlm.nih.gov/search (accessed on 20 October 2025). In 2025, the GeneCards database https://www.genecards.org/ (accessed on 20 October 2025), OMIM http://omim.org/ (accessed on 20 October 2025), and TTD https://db.idrblab.net/ttd/ (accessed on 20 October 2025) databases were consulted, with ‘Neurodegenerative diseases’ entered as the search term and the search species set to ‘Homo sapiens.’ Standard gene names were obtained from UniProt https://www.uniprot.org/ (accessed on 20 October 2025), and all predicted gene entries from the three databases were merged, with duplicate entries removed.

### 4.3. Intersection of Toxicological Targets and Disease Targets

JQuery tools (v1.6.x) https://jvenn.toulouse.inrae.fr/app/index.html (accessed on 20 October 2025) were used to overlap the predicted targets of 2,4-D and targets related to neurodegenerative diseases to identify cross-targets. Cross-targets were further analysed.

### 4.4. Protein–Protein Interaction (PPI) Network Construction

The intersecting targets obtained from the Venn diagram were imported into the String database https://string-db.org (accessed on 20 October 2025) to construct a protein–protein interaction (PPI) network. The network type was set to ‘Full Network,’ the species was set to Homo sapiens, and interactions with scores greater than 0.4 were retained, while disconnected nodes were removed from the network. The TSV file exported from STRING was then imported into Cytoscape 3.10.0 for visualisation and analysis. The cytoNCA plugin was used for analysis and scoring, and key nodes were comprehensively evaluated based on three centrality metrics: Betweenness, Closeness, and Degree. The core targets were selected based on the following criteria: Degree ≥ 2, Closeness ≥ 0.28, and Betweenness ≥ 0. Degree represents the number of connections between a node and other nodes in the network, measuring the node’s direct influence. Nodes with high degrees are typically ‘core members’ of the network. On the other hand, Betweenness measures the frequency with which a node appears in the shortest paths between all other node pairs. A higher value indicates that the node plays a greater ‘bridge’ or ‘hub’ role in the network. Closeness reflects the inverse of the average shortest path length from the node to all other nodes. A higher value indicates that the node is closer to the network centre. The central nodes of the PPI network were identified, and the top 12 core targets were screened based on the comprehensive ranking.

### 4.5. GO and KEGG Pathway Enrichment Analysis

The intersection targets obtained from the previous network toxicology were uploaded to the SRplot platform https://www.bioinformatics.com.cn/srplot (accessed on 20 October 2025), selecting the species Homo sapiens. Gene Ontology (GO) enrichment analysis was performed, with categories including Biological Process (BP), Molecular Function (MF), and Cellular Component (CC), setting *p* < 0.05. The top 10 enriched entries were selected for the enrichment bubble plot, with the vertical axis representing the entry name, bubble size representing the number of genes, and colour mapping the *p*-value. The top 10 KEGG pathways were screened with significantly enriched target genes based on the lowest *p*-value, with a screening criterion of *p* < 0.05. A Sankey–Bubble hybrid plot was created, with the left side being a Sankey diagram representing the genes contained in each pathway, and the right side being a conventional bubble plot, where bubble size indicates the number of genes in the pathway and bubble colour indicates the *p*-value.

### 4.6. Molecular Docking

The core targets selected based on the three centrality metrics of Betweenness, Closeness, and Degree were subjected to molecular docking analysis with 2,4-D to assess the potential interactions between 2,4-D and the core targets. The core targets were designated as receptors, while 2,4-D served as the ligand for docking validation. The three-dimensional structure of 2,4-D was obtained from the PubChem database https://pubchem.ncbi.nlm.nih.gov (accessed on 22 October 2025) in SDF format. The crystal structures of the core targets were obtained from the RCSB Protein Database https://www.rcsb.org (accessed on 22 October 2025), with the species set to ‘Homo sapiens’ and refinement resolution ≤ 2.5, and the PDB files were downloaded. The Discovery Studio 4.5 Client software was used to preprocess the small-molecule ligands and target proteins, removing water molecules, potential cofactors, and unnecessary ligands, and adding polar hydrogen bonds. Molecular docking was performed using the CB-Dock2 platform https://cadd.labshare.cn/cb-dock2/php/index.php (accessed on 22 October 2025). The docking site with the lowest Vina score was selected as the optimal binding mode. The binding energy was calculated, and a three-dimensional visualisation of the binding mode was generated. CB-Dock2 was selected for molecular docking due to its capability of automatic binding site prediction, which helps reduce operational bias and is suitable for multi-target analysis. The tool has demonstrated reliable performance in benchmark tests. However, docking scores are theoretical estimates derived from scoring functions and may not fully account for key biophysical factors such as solvation effects and protein flexibility. Thus, the docking results should be interpreted as computational inferences, requiring further contextual analysis for comprehensive understanding.

## 5. Conclusions

This study systematically investigated the potential mechanisms underlying the association between the herbicide 2,4-D and neurodegenerative diseases (NDDs) by integrating network toxicology and molecular docking techniques. The findings revealed that 2,4-D may induce neuronal damage by interfering with multiple key targets and pathways, thereby increasing the risk of NDDs. The study identified 89 common targets between 2,4-D and NDDs, among which 12 core targets were characterised. These targets play crucial roles in pathological processes such as inflammatory responses, metabolic disorders, oxidative stress, vascular damage, and synaptic dysfunction. Molecular docking analysis confirmed that 2,4-D binds most significantly to the core targets NFKB1 and NOS3. Functional enrichment analysis revealed that 2,4-D may influence neural system homeostasis through 10 key pathways. These findings provide a molecular-level explanation for the neurotoxic mechanisms of 2,4-D and its potential role in neurodegenerative diseases.

## Figures and Tables

**Figure 1 ijms-26-11980-f001:**
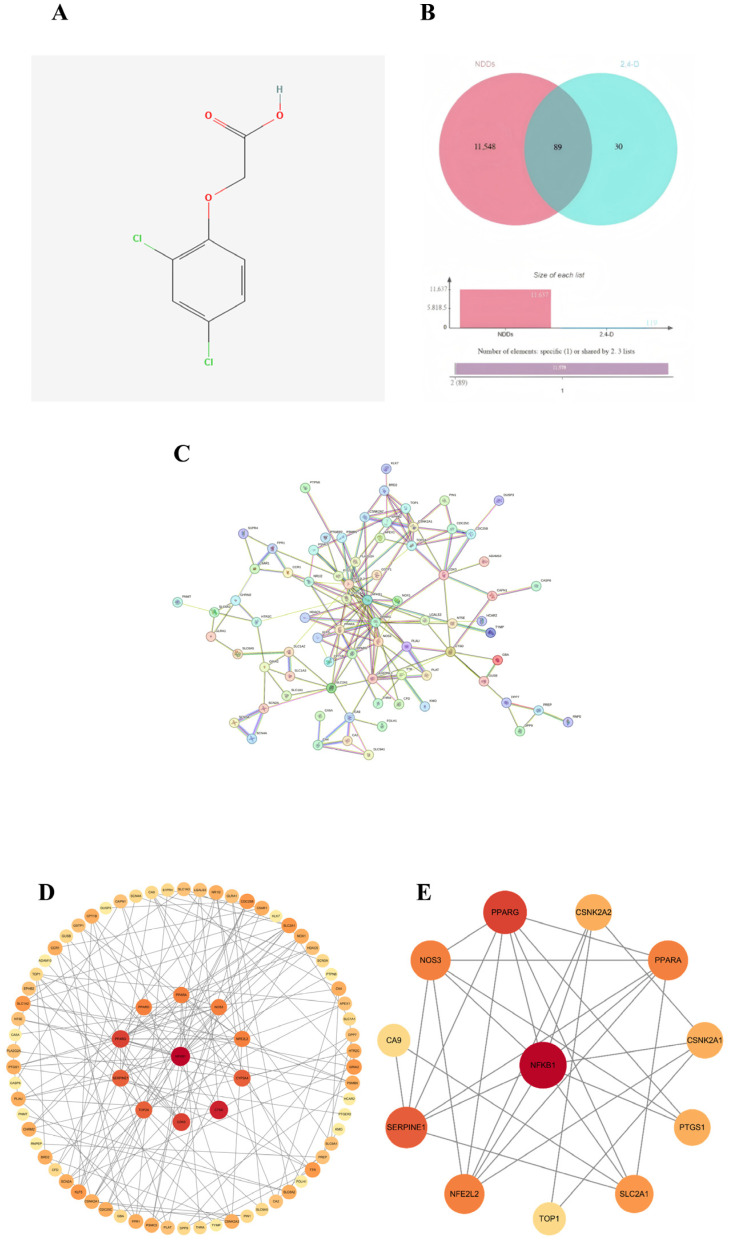
(**A**) Two-dimensional structure diagram of 2,4-D; (**B**) Venn diagram showing the overlap between 2,4-D and target genes of neurodegenerative diseases; (**C**) STRING database construction of a single protein–protein interaction (PPI) network; (**D**) PPI network diagram of core targets (circular); (**E**) the top 12 hub nodes with the highest degree were identified; the network diagram illustrates the complex interactions among the 12 core genes, highlighting their functional connections. Among these, NFKB1, PPARG, SERPINE1, PPARA, NOS3, NFE2L2, SLC2A1, CSNK2A2, CSNK2A1, PTGS1, TOP1, and CA9 exhibit the highest MCC values, represented by darker red node colours in the figure.

**Figure 2 ijms-26-11980-f002:**
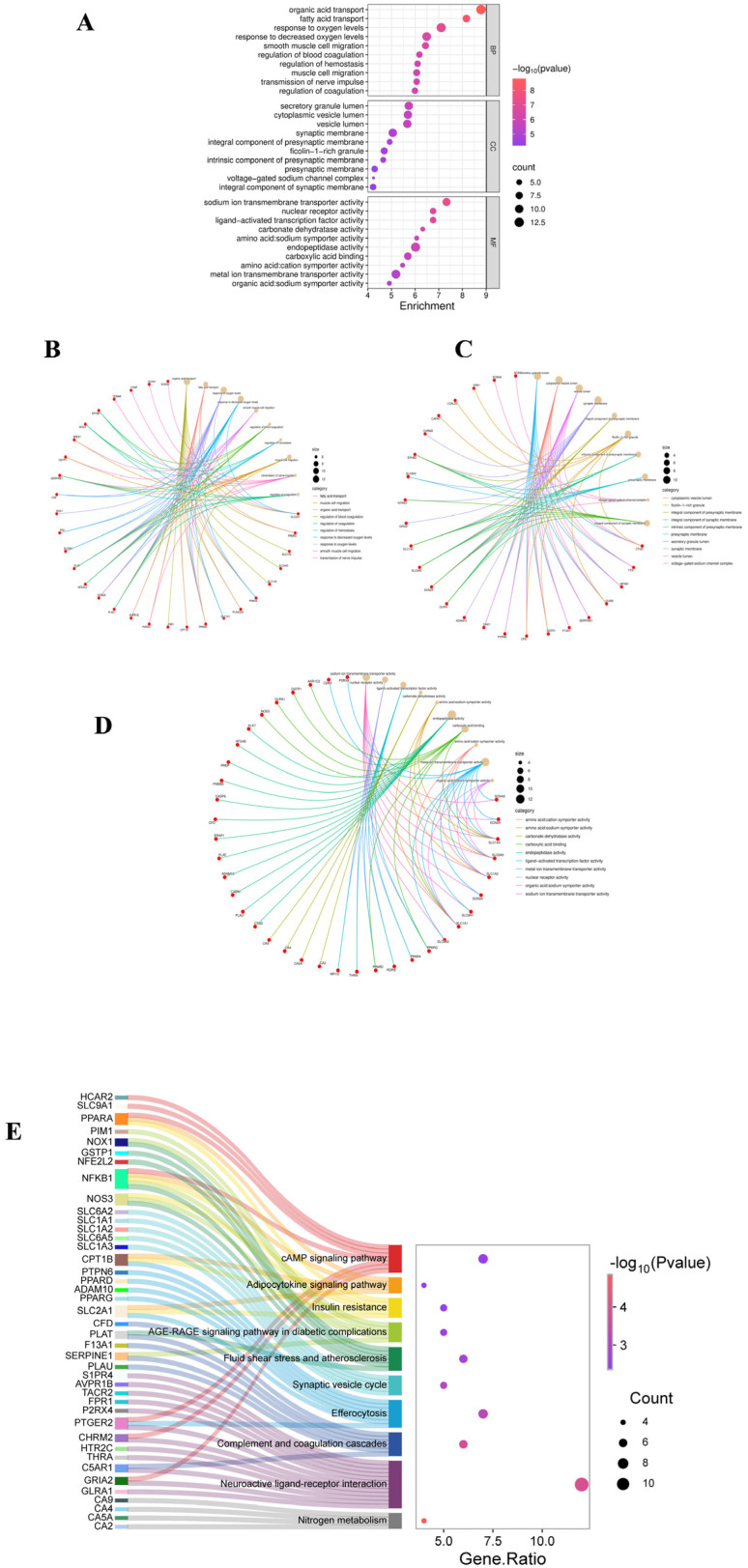
(**A**) GO analysis bubble chart (BP, CC, MF); (**B**) GO analysis chart of BP; (**C**) GO analysis chart of CC; (**D**) GO analysis chart of MF; (**E**) The KEGG Sangi bubble plot displays the relationship between the top-ranked target genes and the top 10 pathways in gene proportion order.

**Figure 3 ijms-26-11980-f003:**
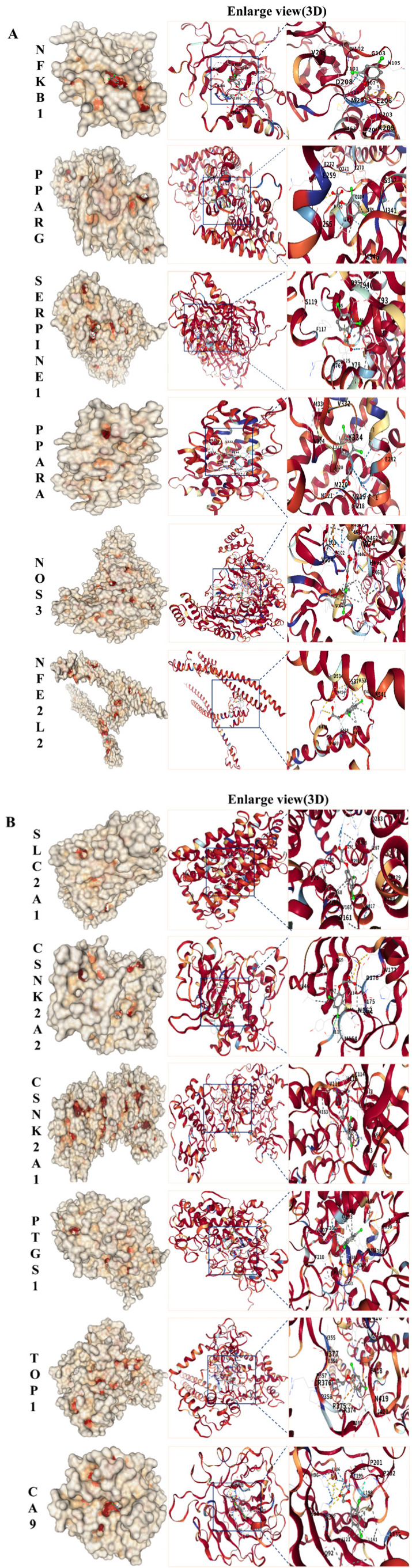
(**A**) Molecular docking results of 2,4-D with NFKB1, PPARG, SERPINE1, PPARA, NOS3, and NFE2L2, all showing the lowest binding energy. (**B**) Molecular docking results of 2,4-D with SLC2A1, CSNK2A2, CSNK2A1, PTGS1, TOP1, and CA9, all showing the lowest binding energy.

**Figure 4 ijms-26-11980-f004:**
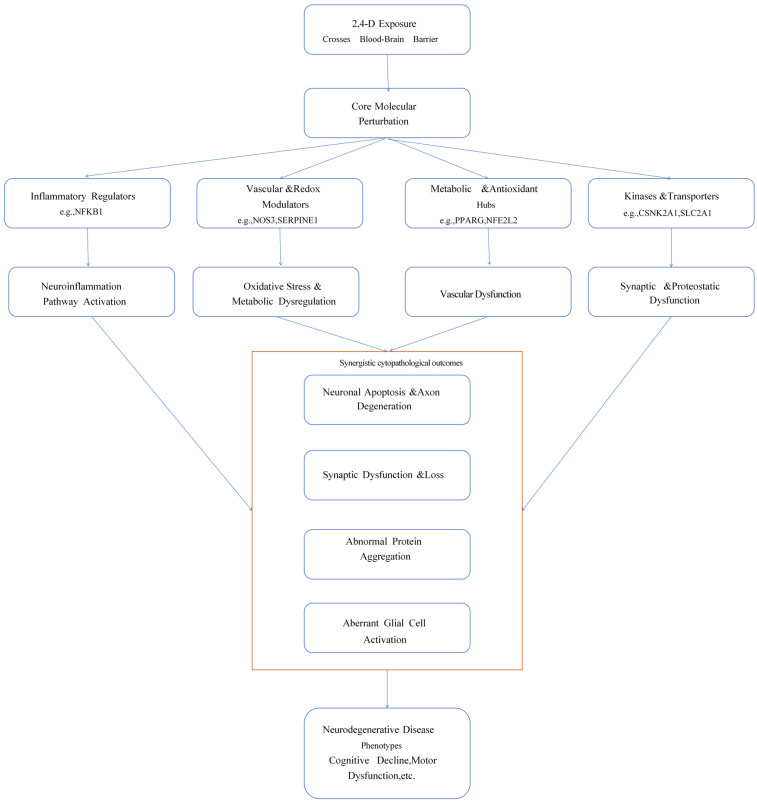
Schematic diagram of the potential multi-target multi-pathway mechanism of 2,4-D induced neurodegenerative diseases.

**Table 1 ijms-26-11980-t001:** The result of molecular docking.

Ingredient	Target	Binding Energy (kcal⋅mol^−1^)
2,4-Dichlorophenoxyacetic acid	NFKB1	−5.1
2,4-Dichlorophenoxyacetic acid	PPARG	−5.9
2,4-Dichlorophenoxyacetic acid	SERPINE1	−6.2
2,4-Dichlorophenoxyacetic acid	PPARA	−6.1
2,4-Dichlorophenoxyacetic acid	NOS3	−7.1
2,4-Dichlorophenoxyacetic acid	NFE2L2	−5.2
2,4-Dichlorophenoxyacetic acid	SLC2A1	−6.0
2,4-Dichlorophenoxyacetic acid	CSNK2A2	−6.6
2,4-Dichlorophenoxyacetic acid	CSNK2A1	−6.5
2,4-Dichlorophenoxyacetic acid	PTGS1	−6.8
2,4-Dichlorophenoxyacetic acid	TOP1	−5.7
2,4-Dichlorophenoxyacetic acid	CA9	−6.4

**Table 2 ijms-26-11980-t002:** Summary of core targets, associated pathways, and their biological roles.

Core Target	Associated Pathways (KEGG)	Core Biological/Pathological Role
NFKB1	Neuroactive ligand–receptor interaction; AGE-RAGE signaling pathway in diabetic complications	Inflammatory transcriptional regulator
PPARG	Insulin resistance; Adipocytokine signaling pathway; cAMP signaling pathway	Nuclear receptor for metabolism and inflammation regulation
SERPINE1	Complement and coagulation cascades; Fluid shear stress and atherosclerosis	Inhibitor of the fibrinolytic system
PPARA	Insulin resistance; Adipocytokine signaling pathway	Nuclear receptor for fatty acid metabolism regulation
NOS3	Nitrogen metabolism; Fluid shear stress and atherosclerosis	Nitric oxide synthase (endothelial)
NFE2L2	Efferocytosis; AGE-RAGE signaling pathway in diabetic complications	Master regulator of antioxidant response
SLC2A1	cAMP signaling pathway	Glucose transporter (GLUT1)
CSNK2A1	Synaptic vesicle cycle; cAMP signaling pathway	Serine/threonine-protein kinase
CSNK2A2	Synaptic vesicle cycle; cAMP signaling pathway	Serine/threonine-protein kinase subunit
PTGS1	Complement and coagulation cascades; Fluid shear stress and atherosclerosis	Prostaglandin synthase (COX-1)
TOP1	Efferocytosis	DNA topoisomerase I
CA9	Nitrogen metabolism; Fluid shear stress and atherosclerosis	Carbonic anhydrase IX

## Data Availability

The original contributions presented in this study are included in the article. Further inquiries can be directed to the corresponding authors.

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
