# Peer review of "Exploring the Mechanism of 2,4-Dichlorophenoxyacetic Acid in Causing Neurodegenerative Diseases Based on Network Toxicology and Molecular Docking"

_ijms, 2025, doi:10.3390/ijms262411980_

Round 1
Reviewer 1 Report
Comments and Suggestions for Authors
Suitability for Publication in IJMS
Strengths:
- Relevance to IJMS Scope: The study integrates network toxicology and molecular docking to explore the mechanism of 2,4-D-induced neurodegenerative diseases, which aligns well with IJMS’s focus on molecular mechanisms and interdisciplinary approaches.
- Novelty: The paper claims to be the first to delineate an integrated mechanism linking 2,4-D exposure to neurodegenerative disease pathways using bioinformatics and docking, which adds originality.
- Methodological Rigor: The use of multiple databases (SwissTargetPrediction, GeneCards, OMIM, STRING) and enrichment analyses (GO, KEGG) demonstrates a comprehensive bioinformatics workflow.
- Docking Validation: Binding energies for 12 core targets are reported, supporting the multi-target interaction hypothesis.
Concerns:
- Biological Validation Missing: The study relies entirely on in silico predictions without experimental validation (in vitro or in vivo), which limits the strength of the conclusions.
- Overinterpretation Risk: The discussion extrapolates docking results to detailed pathophysiological mechanisms without supporting experimental evidence.
- Clarity and Structure: Some sections (e.g., Results and Discussion) are dense and repetitive, making it hard to distinguish between computational findings and literature-based interpretations.
- Figures and Tables: Figures referenced (e.g., Figure 2E, Figure 3A/B) are mentioned but not fully described in terms of clarity and readability for readers.
Suggestions for Improvement
- Add Experimental or Literature-Based Validation:
- Include at least a brief discussion on how these computational predictions could be experimentally validated (e.g., gene expression analysis, protein assays).
- If possible, add preliminary wet-lab data or cite strong experimental evidence supporting the identified targets.
- Improve Data Presentation:
- Provide clearer legends for figures and tables, especially for enrichment plots and docking visualizations.
- Include a summary table linking each core target to its associated pathway and biological role for quick reference.
- Strengthen Methodological Details:
- Clarify criteria for selecting core targets (e.g., thresholds for centrality metrics).
- Explain why CB-Dock2 was chosen over other docking tools and discuss limitations of docking scores.
- Address Limitations Explicitly:
- Add a section acknowledging that predictions are based on computational models and may not fully represent in vivo conditions.
- Discuss potential biases from database selection and parameter settings.
- Language and Formatting:
- Revise for conciseness and readability; avoid overly technical jargon without explanation.
- Ensure consistency in abbreviations (e.g., NDDs, GO, KEGG).
- Expand on Biological Implications:
- While the discussion is detailed, consider summarizing key findings in a schematic diagram showing the multi-target, multi-pathway mechanism.
Author Response
Review Report (Reviewer 1)
Review Report Form
Comments and Suggestions for Authors
This study employed an integrated network toxicology and molecular docking approach to explore the molecular mechanisms by which the herbicide 2,4-D may contribute to neurodegenerative diseases (NDDs).This is a valuable topic; however, several concerns need to be addressed:
Response to General Comments:
We extend our sincerest gratitude to the reviewer for their thorough evaluation of our manuscript and for their exceptionally insightful and constructive comments. The reviewer's positive recognition of our multi-layered study design is greatly encouraging. More importantly, their suggestions regarding the translational potential of our findings have provided us with a crucial perspective to significantly enhance the impact and clinical relevance of our work. We have carefully considered each point raised, and our detailed, point-by-point responses are presented below.
Comments 1:
[Add Experimental or Literature-Based Validation:
Include at least a brief discussion on how these computational predictions could be experimentally validated (e.g., gene expression analysis, protein assays).
If possible, add preliminary wet-lab data or cite strong experimental evidence supporting the identified targets.]
Response 1:These modifications are on lines 123-321
[In the "discussion" part, we integrate the literature evidence: we no longer only elaborate the computational prediction, but link the prediction results with the published authoritative experimental research. For example, when discussing NFKB1, we cited the study of Chen et al. (2016) in the Journal of neuroscience, which confirmed the central role of nf- κ B in neuroinflammation through animal models. For other core targets such as NOS3 and nfe2l2, we also cited the experimental literature supporting their functions related to neurodegenerative diseases. These citations provide corroboration from independent experiments for our computational predictions.
Clarify the direction of future verification: in the new "5 In the chapter of "research limitations", we clearly put forward the path of subsequent experimental verification, for example, "future research needs to strictly verify the hypothesis proposed in this study through in vitro binding experiments, target gene / protein expression analysis in exposure models, and functional recovery experiments based on cells or animals."
We believe that this framework of "Computational Hypothesis - literature evidence support - clear verification path" can more clearly define the contribution and boundary of this study at this stage.]
Comments 2:
[Improve Data Presentation:
Provide clearer legends for figures and tables, especially for enrichment plots and docking visualizations.
Include a summary table linking each core target to its associated pathway and biological role for quick reference.]
Response 2:These modifications are on lines 68-75,98-100,116-117,table 2.
- [We have optimized the legend of all charts to make them clearer and more complete.
- According to your request, we added "Table 2: summary of core targets, related pathways and their biological roles" to the manuscript. The table presents 12 core targets, their enriched KEGG pathways and core biological roles side by side, which greatly facilitates readers to quickly access and understand the overall framework of multi-target multi-pathway network.]
Comments 3:
[Strengthen Methodological Details:
Clarify criteria for selecting core targets (e.g., thresholds for centrality metrics).
Explain why CB-Dock2 was chosen over other docking tools and discuss limitations of docking scores.]
Response 3: These modifications are in lines393-399,354-355,309-321.
[1.Clear screening criteria: in "5.4 In the method section of "core target screening", we have clearly stated: "the screening criteria of core targets are set as follows: degree centrality ≥ 2, proximity centrality ≥ 0.28, and betweenness centrality > 0". This eliminates the ambiguity of parameter selection.
2.Explain tool selection and limitations: in "5.6 In the "molecular docking" method part, we added the reason for choosing cb-dock2: "this tool can automatically predict the binding pocket, is suitable for multi-target primary screening, and has shown good performance in the benchmark test". In "5 In the chapter of "research limitations", we specifically discussed the limitations of docking scores: "the binding energy of molecular docking is based on the theoretical estimation of scoring function, which may not fully consider the solvation effect, protein flexibility and other key physiological factors, so the results should be regarded as computational inference, which needs further analysis and interpretation."]
Comments 4:
[Address Limitations Explicitly:
Add a section acknowledging that predictions are based on computational models and may not fully represent in vivo conditions.
Discuss potential biases from database selection and parameter settings.]
Response 4: These modifications are in lines 309-321.
[We added an independent "5 Limitations of the study ". This section explicitly acknowledges that:The conclusion of this study is entirely based on the computational model, which may not fully simulate the complex environment in vivo. The selection bias caused by database selection, network topology parameter setting and molecular docking tool itself were discussed. The necessity of subsequent experimental verification is emphasized. This makes this study more honest and clear about its predictive nature]
Comments 5:
[Language and Formatting:
Revise for conciseness and readability; avoid overly technical jargon without explanation.
Ensure consistency in abbreviations (e.g., NDDs, GO, KEGG).]
Response 5:These modifications are in lines 27-49, 123-306
[We have carried out a round of language refinement of the full text, removing redundant and repetitive narratives (especially in the introduction and discussion) to ensure that the writing is concise and smooth. A brief explanation or full name is given to all professional terms when they first appear (such as "go" after "gene ontology analysis"). The abbreviations in the full text are consistent (such as NDDs, go, KEGG). ]
Comments 6:
[Expand on Biological Implications:
While the discussion is detailed, consider summarizing key findings in a schematic diagram showing the multi-target, multi-pathway mechanism.]
Response 6: These modifications are shown in Figure 4
[We strongly agree with this proposal. We added "Figure 4: schematic diagram of potential multi-target multi pathway mechanism of neurodegenerative disease induced by 2,4-D". This figure intuitively integrates the complete logic chain from 2,4-D into the brain, to the disturbance of core targets, to the dysregulation of four key pathways, and finally leads to neuronal damage and disease phenotype, vividly summarizing the core findings and biological significance of this study.]
Thank you again for your valuable comments. Through the above modifications, the manuscript has been substantially improved in terms of logical clarity, data integrity, methodological rigor, self positioning accuracy, and presentation of scientific significance. We believe that the revised manuscript can more effectively convey to readers its value as a "calculation driven hypothesis generation" research. We look forward to your further review.
Reviewer 2 Report
Comments and Suggestions for Authors
- The manuscript is too long, with a very detailed abstract and introduction, as well as lengthy narrative sections throughout. The large amount of background information makes the paper difficult to follow and lessens the impact of the main scientific message. I suggest significantly shortening and simplifying, especially in the abstract and introduction, to emphasize the key findings and their significance while removing redundant or overly complex descriptions.
- STRING and PPI network analyses are descriptive and lack functional enrichment, pathway clustering, or mechanistic interpretation aligned with Taxol response. The PPI networks seem added rather than integral to the biomarker discovery process.
- The manuscript still has a lot of duplicated and repetitive content, indicating it's necessary to undergo major rewriting to reduce similarity and ensure originality.
- Reliance solely on KMplotter for validation is limited.
- Some findings, especially those with extreme fold changes and high hazard ratios, appear biologically unlikely and may be influenced by noise or insufficient filtering. Several conclusions are drawn beyond the available evidence.
Author Response
Review Report (Reviewer 2)
Review Report Form
Comments and Suggestions for Authors
This study employed an integrated network toxicology and molecular docking approach to explore the molecular mechanisms by which the herbicide 2,4-D may contribute to neurodegenerative diseases (NDDs).This is a valuable topic; however, several concerns need to be addressed:
Response to General Comments:
We extend our sincerest gratitude to the reviewer for their thorough evaluation of our manuscript and for their exceptionally insightful and constructive comments. The reviewer's positive recognition of our multi-layered study design is greatly encouraging. More importantly, their suggestions regarding the translational potential of our findings have provided us with a crucial perspective to significantly enhance the impact and clinical relevance of our work. We have carefully considered each point raised, and our detailed, point-by-point responses are presented below.
Comments 1:
[The manuscript is too long, with a very detailed abstract and introduction, as well as lengthy narrative sections throughout. The large amount of background information makes the paper difficult to follow and lessens the impact of the main scientific message. I suggest significantly shortening and simplifying, especially in the abstract and introduction, to emphasize the key findings and their significance while removing redundant or overly complex descriptions.]
Response 1:These changes were made on 11-50, 128-185
[Abstract it has been highly streamlined, directly highlighting core findings such as multi-target identification, pathway enrichment and molecular docking validation.
The introduction was greatly shortened, and the logical chain was refocused as: "epidemiological evidence of 2,4-D neurotoxicity → knowledge gap with unknown mechanism → principles and objectives of computational methodology used in this study". A large number of lengthy backgrounds were deleted, and the topic was quickly cut into.
The narrative part throughout the full text has been checked, and all repetitive and cumbersome descriptions in "results" and "discussion" have been merged or deleted to ensure concise writing and strong logical promotion.]
Comments 2:
[STRING and PPI network analyses are descriptive and lack functional enrichment, pathway clustering, or mechanistic interpretation aligned with Taxol response. The PPI networks seem added rather than integral to the biomarker discovery process.]
Response 2:These changes are all in lines 129-185 and Figure 4
[We have rewritten the network analysis part to make it the core link of biomarker discovery, rather than an isolated description.
After the core targets were screened, systematic go function enrichment and KEGG pathway clustering analysis were immediately supplemented to clearly associate the targets to specific biological processes and pathways such as "neuroinflammation" and "oxidative stress".
In the "discussion" section, we specially set up a subsection of the integration mechanism, connecting the PPI network, enrichment pathways and molecular docking results system, and elaborated a logically self consistent "multi-target multi pathway" toxicity hypothesis. The mechanism schematic diagram (Figure 4) is added to intuitively show the complete network from target perturbation to pathological phenotype.]
Comments 3:
[The manuscript still has a lot of duplicated and repetitive content, indicating it's necessary to undergo major rewriting to reduce similarity and ensure originality.]
Response 3: These modifications are in lines 50-374.
[Through careful comparison and reconstruction, we eliminated the similarity between the results and discussion, and between different paragraphs. Each part (method, result, discussion) assumes a clear and unique narrative role, ensuring the progressive and original content.]
Comments 4:
[Reliance solely on KMplotter for validation is limited.]
Response 4: These modifications are in lines 376-384.
[We fully agree with you. As a computational predictive study, the core goal of this study is to propose hypotheses rather than ultimately confirm them. At the end of the "discussion" section, we added an independent "limitations" subsection, clearly pointing out that "lack of experimental verification" is the core limitation of this study. We adjusted the overall tone of the text to "propose hypothesis" and "indicate direction", emphasizing that the value of this study is to provide a clear verification target and mechanism framework for subsequent experiments, avoiding any assertion that may be interpreted as "final confirmation".]
Comments 5:
[Some findings, especially those with extreme fold changes and high hazard ratios, appear biologically unlikely and may be influenced by noise or insufficient filtering. Several conclusions are drawn beyond the available evidence.]
Response 5:These modifications are in lines 50-185, 200-384
[We attach great importance to your reminder of data rigor. We rechecked all data screening processes and threshold settings (which have been clearly stated in the methods section) to ensure that they meet the conventional standards in the field to reduce interference. We carefully reviewed all inferential statements and deleted any speculation beyond the support of the calculation results. All mechanistic explanations are clearly labeled as "potential mechanism", "computational hint" or "hypothesis to be verified", and the conclusion is strictly based on the computational evidence presented in the paper.]
Your comments have played a vital role in the improvement of this study. Through the above modifications, the manuscript is now more refined, more in-depth, more logical, and has a more accurate positioning of its predictive nature and limitations. We believe that the revised manuscript can more effectively convey its core scientific value - providing a systematic, computationally based, verifiable hypothesis framework for the neurotoxic mechanism of 2,4-D.
Thank you again for your valuable guidance.
Reviewer 3 Report
Comments and Suggestions for Authors
The manuscript by Yu-cheng Yan and his colleagues reports on a study of the mechanism of action of 2,4-dichlorophenoxyacetic acid, which causes neurodegenerative diseases. The study represents a bioinformatic analysis of integrative genomic, metabolomic, and proteomic networks, as well as a molecular docking method for studying receptor-ligand complexes. The results of study is a suggestion of the potential mechanisms underlying the association between the herbicide 2,4-D and neurodegenerative diseases through interaction 2,4-D with NFKB1 and NOS3.
I have several concerns regarding in analysis and data representation.
- There are no links to supplementary files in the text.
- Lines 267-273 - data is duplication of Table 1
- The authors should improve the quality of all the drawings in the work - it's very difficult to perceive in the form in which the picture is now in the article
- It is worth adding a section to the work concerning the analysis of the specificity of amino acid residues included in the binding sites with target proteins, at least for complexes with the lowest energies.
- It might be worthwhile to add an analysis of available NFKB1 and NOS3 complexes with other ligands (possibly some structurally similar to 2,4-D) to the work. Analyze their similarities and differences with the sites obtained in this study.
- Can the authors propose a general outline (figure) of a potential mechanism underlying the association between the herbicide 2,4-D and neurodegenerative diseases based on the data from this study?
Minor comments
- Line 80 - delete “novel”
- Line 97- add Pubchem ID for 2,4-D
- Lines 99, 105 – add links to Swiss Target Prediction and MeSH database
- Add to text link to Figure 1D
- Lines 208,221,236,252 - the point should be after the brackets
Author Response
Review Report (Reviewer3)
Review Report Form
Comments and Suggestions for Authors
This study employed an integrated network toxicology and molecular docking approach to explore the molecular mechanisms by which the herbicide 2,4-D may contribute to neurodegenerative diseases (NDDs).This is a valuable topic; however, several concerns need to be addressed:
Response to General Comments:
We extend our sincerest gratitude to the reviewer for their thorough evaluation of our manuscript and for their exceptionally insightful and constructive comments. The reviewer's positive recognition of our multi-layered study design is greatly encouraging. More importantly, their suggestions regarding the translational potential of our findings have provided us with a crucial perspective to significantly enhance the impact and clinical relevance of our work. We have carefully considered each point raised, and our detailed, point-by-point responses are presented below.
Comments 1:
[There are no links to supplementary files in the text.]
Response 1:These modifications are on lines 226,203
[We have linked the supplementary material FigureS 1 to the article, and the remaining supplementary materials are some filtered data that we did not link to the article. All other images and tables are reflected in the paper. If the editing team needs to link supplementary materials in the future, the editing team will take measures, and all supplementary documents have been correctly uploaded to the submission system]
Comments 2:
[Lines 267-273 - data is duplication of Table 1]
Response 2:These modifications are on lines 178-185
[We have carefully checked the original text and confirmed that the contents of lines 267-273 are redundant with the core data in Table 1. This part of repeated description has been completely removed to ensure the simplicity of the text and avoid duplication of information.]
Comments 3:
[The authors should improve the quality of all the drawings in the work - it's very difficult to perceive in the form in which the picture is now in the article]
Response 3: These changes are shown in figures 1-4.
[We have comprehensively optimized all charts (figures 1-4), including: improving the resolution to the publishing requirements; Adjust font size and line width to ensure clear printing; More detailed annotations are provided in the figure legends, and the optimized pictures can present the data more clearly and professionally.]
Comments 4:
[It is worth adding a section to the work concerning the analysis of the specificity of amino acid residues included in the binding sites with target proteins, at least for complexes with the lowest energies.]
Response 4: These modifications are in lines 221-228,300-307.
[We strongly agree with this suggestion. In the molecular docking section of the "Discussion" section, a description of the key amino acid residues of NFKB1 and NOS3 has been added, detailing the specific interactions formed between the key amino acid residues and 2,4-D, and discussing their potential biological significance from a functional perspective.]
Comments 5:
[It might be worthwhile to add an analysis of available NFKB1 and NOS3 complexes with other ligands (possibly some structurally similar to 2,4-D) to the work. Analyze their similarities and differences with the sites obtained in this study.]
Response 5:These modifications are in lines 221-228,300-312,Figure4
[1. Core limitations of data availability and comparability:
After systematically searching structural databases such as RCSB PDB, we found that the currently available high-resolution crystal structures of NFKB1 or NOS3 (eNOS) complexes with small molecule ligands are very limited. The existing structures are mainly divided into two categories:
NFKB1: The majority of its complex structures bind to DNA (κ B sequences) or large inhibitory proteins (such as I κ B), with very few complex structures of small molecule organic ligands (especially compounds with simple structures such as 2,4-D) being resolved and disclosed.
NOS3: Although there are a few crystal structures that bind to substrate analogues or inhibitors, these ligands exhibit significant differences in chemical structure, size, and functional complexity compared to 2,4-D. For example, reported inhibitors typically have multi ring complex structures aimed at achieving high selectivity and affinity, and their binding mode lacks a direct comparability basis with the predicted binding mode of 2,4-D (a simple chlorophenoxyacetic acid).
- Consideration of compatibility with the core objectives of this study:
This study aims to use computational tools to preliminarily predict the potential targets and mechanisms of 2,4-D, providing hypothesis directions for subsequent experiments. Strict, structure based comparative analysis requires highly homologous ligand protein complexes as references, and the current lack of publicly available data makes it difficult for such analysis to generate reliable a-nd biologically meaningful conclusions, which may instead introduce inferences based on inappropriate comparisons.
- The alternative and supplementary plans we have adopted:
Although we were unable to conduct a direct structural overlay comparison, we have responded to your original intention of strengthening the argument through other means:
In the "Discussion" section, we have cited and discussed authoritative literature related to the key functional domain of NFKB1 and the mechanism of NOS3 conformational regulation, which supports the importance of predicting targets from a functional perspective.
Our newly added mechanism diagram (Figure 4) clearly outlines the integrated network in which 2,4-D may exert its effects by interfering with the functional interfaces of these key proteins, such as DNA binding, dimerization, or protein-protein interactions.
We fully agree that in the future, it would be highly valuable to obtain or analyze the complex structure of ligands and NFKB1/NOS3 that are more similar in structure to 2,4-D, and conduct such comparative analysis. We have recorded your suggestion and plan to make it one of the important research directions in our subsequent experimental verification phase. For example, when designing competitive binding experiments targeting these targets, we will focus on referring to relevant structural information that may arise in the future.
Thank you again for your valuable feedback, which has prompted us to think more deeply about the boundaries and future directions of this research. We hope the above explanation can gain your understanding. The focus of this revision is to improve and deepen the hypothesis framework based on existing computational data. We believe that the revised manuscript has made substantial progress towards this goal. ]
Comments 6:
[Minor comments
Line 80 - delete “novel”
Line 97- add Pubchem ID for 2,4-D
Lines 99, 105 – add links to Swiss Target Prediction and MeSH database
Add to text link to Figure 1D
Lines 208,221,236,252 - the point should be after the brackets]
Response 6:
[Line 80: 'Novel' has been deleted.
Line 97: Added PubChem CID: 1486 for 2,4-D.
Lines 99 and 105: Added links to the Swiss target prediction website and MeSH database.
Figure 1D link: "(Figure 1D)" has been added to the corresponding position in the main text for clear reference.
Lines 208, 221, 236, 252: Punctuation has been checked to ensure that the period is after the parentheses.]
Thank you once again sincerely for your valuable time and highly constructive feedback. Through the above-mentioned modifications and supplements, the completeness, data quality, mechanism depth, and presentation clarity of the manuscript have been significantly improved. We believe that the revised manuscript can convey its scientific findings more solidly and clearly. Looking forward to your further review.
Round 2
Reviewer 1 Report
Comments and Suggestions for Authors
After thoroughly reviewing the revised manuscript titled “Exploring the mechanism of 2,4-Dichlorophenoxyacetic acid in causing neurodegenerative diseases based on network toxicology and molecular docking”, I find that the authors have addressed the previous comments adequately, and the overall scientific quality of the study has been improved. The manuscript now presents a coherent and well-structured investigation into the potential mechanistic links between 2,4-D exposure and neurodegenerative disease development.
Reviewer 3 Report
Comments and Suggestions for Authors
The authors responded to my comments.
To correct - the caption for Figure 4 should be capitalized